# Serum Neutralizing and Enhancing Effects on African Swine Fever Virus Infectivity in Adherent Pig PBMC

**DOI:** 10.3390/v14061249

**Published:** 2022-06-09

**Authors:** Jessica A. Canter, Theresa Aponte, Elizabeth Ramirez-Medina, Sarah Pruitt, Douglas P. Gladue, Manuel V. Borca, James J. Zhu

**Affiliations:** 1Foreign Animal Disease Research Unit, Plum Island Animal Disease Center, Agricultural Research Service, United States Department of Agriculture, Orient, NY 11957, USA; jessica.canter@usda.gov (J.A.C.); apontet22@gmail.com (T.A.); elizabeth.ramirez@usda.gov (E.R.-M.); sarah.pruitt@usda.gov (S.P.); 2Plum Island Animal Disease Center, Oak Ridge Institute for Science and Education, Oak Ridge, TN 37830, USA

**Keywords:** African swine fever virus (ASFV), hyperimmune serum, virus neutralization, flow cytometry, monocyte-derived macrophage, extracellular virions, serum-enhanced virus infection

## Abstract

African swine fever virus (ASFV) causes hemorrhagic fever with mortality rates of up to 100% in domestic pigs. Currently, there are no commercial vaccines for the disease. Only some live-attenuated viruses have been able to protect pigs from ASFV infection. The immune mechanisms involved in the protection are unclear. Immune sera can neutralize ASFV but incompletely. The mechanisms involved are not fully understood. Currently, there is no standardized protocol for ASFV neutralization assays. In this study, a flow cytometry-based ASFV neutralization assay was developed and tested in pig adherent PBMC using a virulent ASFV containing a fluorescent protein gene as a substrate for neutralization. As with previous studies, the percentage of infected macrophages was approximately five time higher than that of infected monocytes, and nearly all infected cells displayed no staining with anti-CD16 antibodies. Sera from naïve pigs and pigs immunized with a live-attenuated ASFV and fully protected against parental virus were used in the assay. The sera displayed incomplete neutralization with MOI-dependent neutralizing efficacies. Extracellular, but not intracellular, virions suspended in naïve serum were more infectious than those in the culture medium, as reported for some enveloped viruses, suggesting a novel mechanism of ASFV infection in macrophages. Both the intracellular and extracellular virions could not be completely neutralized.

## 1. Introduction

African swine fever virus (ASFV) infects wild boars and domestic pigs and causes hemorrhagic fever with mortality rates of up to 100% in the latter [1]. Currently, there is no commercial vaccine for the disease. To date, all experimental inactivated or subunit ASF vaccine candidates have been unable to protect pigs with high efficacy against a highly virulent ASFV strain [2,3]. Several live-attenuated viruses have shown promise as experimental vaccine candidates that offer up to 100% protection. The protective mechanisms involved in these experimental vaccines is not clear even though both humoral and cell-mediated immunity have been shown to be important in the immune protection against ASF [4,5,6,7]. The effects of humoral immunity on ASFV infection have been demonstrated in pigs with passive transfer of colostrum or serum antibodies from convalescent pigs, which reduced viremic titers, sickness and mortality after ASFV challenge [4,8,9]. 

ASFV-specific antibodies showed a virus neutralization effect in plaque reduction assays [10,11,12,13,14,15]; however, even in the presence of high antibody titers, a fraction of the infective virus population remains non-neutralizable [10,12]. It has been found that the content of phospholipids in ASFV particles affected antibody neutralization of low- and high-passage viruses [16], and the presence of non-neutralizing antibodies inhibited the effect of ASFV-neutralizing antibodies [17].

The traditional plaque reduction–based ASFV neutralization assays previously used are time-consuming, requiring 5–7 days, and are very laborious. To speed up the assay, a genetically modified ASFV expressing a chromogenic marker gene was developed [18]; regardless, the manual quantification of infected cells is a tedious process, and the small number of ASFV used in these assays is usually ~100 plaques, which results in large technical variations. Therefore, a need exists to develop a rapid and robust immune assay to characterize protective immunity against ASFV. Flow cytometry has been applied to measure neutralizing effect on vaccinia virus [19] and ASFV infection in the Vero cell line [20,21]. This technique was also used to measure ASFV infectivity in pig macrophages [22,23]. Prior to this report, flow cytometry has not been used in ASFV neutralization assays in pig macrophages. 

We recently developed a virulent, recombinant ASFV24 (ASFV-G-ΔMGF13/14-EGFP) expressing an enhanced green fluorescent protein (EGFP) [24]. EGFP is a common fluorescent protein used as a marker in flow cytometry analysis [25]. This recombinant virus maintains virulence in pigs and grows to equivalent titers in primary macrophage cell cultures as the wild-type ASFV Georgia 2007 isolate [24]. The incorporation of this fluorescent marker allows for easy detection of ASFV-infected cells in vitro using fluorescent microscopy or FACS analysis. This report describes the development of a rapid immunoassay for the quantification of the protective effect provided by ASF immune sera against ASFV infection in pig macrophages. Here, ASFV-G-ΔMGF13/14-EGFP was used in a flow cytometry–based immunoassay to quantify virus neutralization by hyperimmune sera against ASFV infection in swine macrophages. 

## 2. Materials and Methods

### 2.1. Adherent PBMC and Serum Samples

Primary swine adherent peripheral blood mononuclear cells (PBMC) were use in this study, which were isolated from healthy donor pigs, as described previously [26]. In summary, heparinized blood was incubated for 2 h at 37 °C to allow for separation of leukocyte and erythrocyte fractions before layering the leukocyte fraction over a Ficoll–Paque PLUS (Cytivia) density gradient (density of 1.077 g/L). PBMC were separated by centrifugation. The PBMC isolated from this step were cultured overnight at 37 °C with 5% CO_2_ in RPMI 1640 medium (Life Technologies, Carlsbad, CA, USA) supplemented with 30% L929 supernatant, heterologous porcine plasma, 20% fetal bovine serum (characterized and gamma irradiated; HyClone, GE Healthcare, Chicago, IL, USA), antibiotic/antimycotic (Gibco, Waltham, MA, USA) and gentamycin (Gibco) (Complete medium) in Primaria (Falcon) tissue culture flasks. After overnight incubation, non-adherent cells were removed and adherent PBMC were collected with 10 mM EDTA in 1x D-PBS and cultured in 12- or 24-well plates with a macrophage medium [RPMI 1640 medium (Life Technologies) supplemented with 30% L929 supernatant, 20% fetal bovine serum (characterized and gamma irradiated; HyClone, GE Healthcare), antibiotic/antimycotic (Anti/Anti, Gibco) and gentamycin (Gibco)] at 5 × 10^5^ cells per well for 24-well plates or 3 × 10^6^ cells per well for 12-well plates.

Control serum samples were collected from naïve pigs (not vaccinated or challenged) that were donors for blood samples or also used in the experiments of immunization and challenge and named as serum (−). Hyperimmune sera, named as sera (+), were collected from four pigs infected with ASFV-G-ΔI177L and challenged with the parental wild type at 21 days post-challenge, as in the experiment reported by Borca et al. [27]. These pigs did not show any clinical signs of the disease, including viremia after wild-type challenge. The samples were stored in −70 °C until used. Serum samples were not heat-inactivated to include the effect of complements in the assay to test whether the effect improves virus neutralization. The complement effect was compared between removal (without effect of complement-directed cytotoxicity, CDC) and non-removal (with CDC effect) inoculum (the mixture of ASFV and immune serum), as described later.

### 2.2. Virus Production and Titration

Virus used in this study was ASFV-G-ΔMGF13/14-EGFP, which was derived from the parental strain ASFV Georgia 2010 isolate (ASFV-G) and where the MGF13/14 coding sequence was replaced with an EGFP coding sequence, as described elsewhere [24]. The in vitro growth and virulence of this modified virus in pigs are practically the same as the parental virus ASFV-G [24]. Stocks of this virus were cultivated from adherent PBMC in the macrophage medium seeded into Primaria (Falcon) T75 tissue culture flasks for 4 days until nearly 100% of the cultured macrophages exhibited cytopathic effects (CPE). After this incubation, the T75 flasks were frozen at −70 °C for 24 h and then thawed to allow for the collection of the entire culture volume including supernatant and cellular materials. This culture volume was pooled and clarified by centrifugation twice to remove cellular materials. The clarified supernatant was then frozen at −70 °C as stock virus. The virus was titrated with 10-fold serial dilutions in primary swine macrophages seeded into Primaria 96-well plates. Pig erythrocytes from the same donor animal were added to the wells for determination of HAD_50_ based on rosette formation. Viral HAD_50_ titers were calculated using the Reed–Muench method [28].

To test the infectivity of intracellular and extracellular virions, adherent pig PBMC were infected with ASFV at multiplicities of infection (MOI) of 2 for two hours in a culture flask. After 2 h infection, the inoculum was removed and replaced with culture medium. Extracellular ASFV in the culture supernatant was harvested at 15 h post infection, and then the adherent PBMC were washed with the culture medium. After washing, the culture medium was added to the flask, and the adherent PBMC were immediately placed in a −70 °C freezer for three freeze-and-thaw cycles to release intracellular ASFV. The harvested supernatants were centrifugated and filtered through 0.45 μm Spin-X Centrifuge Tube Filter (Costar) to remove cells and debris before storage in the −70 °C freezer. The intracellular and extracellular ASFV were titrated as stated previously.

### 2.3. Virus Infection

To determine the timing of EGFP signal and mature ASFV release into culture medium, swine adherent PBMC were cultured in 12-well plates at 37 °C with 5% CO_2_ infected with ASFV-G-ΔMGF13/14-EGFP at MOI of 1 HAD_50_ calculated based on the number of seeded adherent PBMC. The supernatants and cells were collected in 3 h intervals over the course of 33 h. After supernatant collection, the wells were washed with the culture medium two times and replenished with culture medium for further incubation until the next sampling time. Upon collection, each supernatant sample was centrifuged through a 0.45 µm Spin-X Centrifuge Tube Filter (Costar) to remove all cells in suspension or large debris and the subsequent flow-through was frozen at −70 °C until used. The samples of filtered supernatants (200 μL) were used to inoculate adherent PBMC cultured in a 24-well plate for overnight incubation. The cells in the wells inoculated with the supernatants were collected for flow cytometry analyses using the methods described in Section 2.4. To determine the percentages of infected cells in different MOI, MOI ranging from 0.01 to 4 HAD_50_ were calculated based on the number of adherent PBMC seeded in each well and tested. MOI used in the virus neutralization assays were selected in the range with a linear correlation with the percentages of infected cells in monocyte/macrophage populations. 

### 2.4. Flow Cytometry

Adherent PBMC were collected from culture wells by incubation with 1 mL of 10 mM EDTA in 1x D-PBS after culture medium was removed. Collected cells were fixed in 0.5 mL of FluoroFix (BioLegend, San Diego, CA, USA) before final resuspension in eBioscience Flow Cytometry Staining Buffer (Thermo Fischer, Waltham, MA, USA). To characterize the gated cell populations, anti-pig CD3-FITC (IgG1 clone PPT3), CD14-FITC (IgG2b clone MIL2), CD16-PE (IgG1 clone G7) antibodies (Bio-Rad, Hercules, CA, USA) were used to stain the fixed cells. Each sample was run on a NovoCyte (Agilent Technologies, Santa Clara, CA, USA) for the collection of a minimum of 15,000 events in a volume of 150 µL. The populations of adherent PBMC including monocytes (lower side scatter, SSC) and macrophages (higher SSC) were gated based on the height of SSC (SSC-H) > 0.5 × 10^5^ and height of forward scatter (FSC-H) > 10^5^ as in the study by Fairbairn et al. [29]. ASFV-G-ΔMGF13/14-EGFP was detected as FITC fluorescence using the 488 nm laser 25 and FITC filter, and PE signal was measured with the 488 nm laser and PE filter. EGFP-expressing cells were gated from the populations of monocytes and macrophages by gating EGFP-positive monocytes and macrophages plotted by FITC-H and SSC-H using uninfected samples to set the boundary between EGFP-positive and -negative cells using NovoExpress (Agilent Technologies). For two-color analysis of flow cytometry, cells not stained and cells stained with single fluorescent were used to create the matrixes for compensation between PE and FITC signals. 

### 2.5. Virus Neutralization 

Adherent PBMC were collected, counted and re-seeded in 24-well plates. After >30-h incubation, these cells were infected with an MOI of 0.5 or 0.05 HAD_50_ ASFV (50 μL in culture medium) that was pre-incubated with serum pre-immune or immune serum samples or the culture medium as a negative control for an hour. Each well was inoculated with the mixture (300 μL) of the virus and serial 2-fold diluted serum samples or medium for 2 h. Culture medium (700 μL) was added to each well in the end of infection (without removal of inoculum), or the mixtures were removed and then the wells were added with 1 mL of medium. After overnight (13.5 h) incubation, the cells were collected as stated earlier and analyzed with by flow cytometry, as described earlier. Infected cells were determined as described for flow cytometry using NovoExpress software. The percentage of EGFP+, ASFV-infected cells was calculated by dividing the number of events gated as EGFP+ by the total number of gated monocyte and/or macrophage events excluding cellular debris and lymphocytes as described for flow cytometry. 

### 2.6. Data and Statistical Analysis

The averaged percentages of infected cells in the gated cell populations were calculated from results obtained with individual serum samples from four pigs with duplicates for each sample. For repeated experiments, pooled samples were run in three or four replicates using adherent PBMC prepared from different pigs. The percentages were Log_10_ transformed for statistical analyses with one-tailed *t*-test assuming equal sample variances. Differences with a *p* value of 0.05 or smaller were considered statistically significant. Virus neutralization (VN) was calculated with the formula: VN = (PI_−_ − PI_+_)/PI_−_, where PI_−_ is the percentage of infected cells treated with naïve serum named as serum (−), and PI_+_ is the percentage of infected cells treated with immune serum named as serum (+).

## 3. Results

### 3.1. Characterization of Adherent PBMC by Flow Cytometry

Flow cytometry analyses of non-infected adherent PBMC after overnight culture showed that there were three populations of cells based on count distributions on SSC-H (associated with granularity) and FSC-H (co-related to cell size) plots; one with SSC-H < 2.0 × 10^5^ and FSC-H < 8 × 10^5^ (Population A, ~15%), one with SSC-H < 1.3 × 10^5^ and FSC-H > 8 × 10^5^ (Population B, ~40%) and another with SSC-H > 1.3 × 10^5^ and FCS-H > 8 × 10^5^ (Population C, ~45%) (Appendix A). Population A likely is composed of cell debris and/or platelets based on its lowest FSC-H. Population B is probably lymphocytes, containing ~85% CD3+ cells with high staining intensity (data not shown), ~8% CD14+ and ~7% CD16+ cells with low staining intensity (Appendix A), which shared the same phenotypes as most cells from non-adherent PBMC. Population C is composed of macrophages/monocytes based on its higher averaged SSC-H and FSC-H than those of lymphocytes, only ~15% CD3+ cells (data not shown) and >85% CD14+ and CD16+ cells. Population C can be further divided into two sub-populations as monocytes (SSC-H < 3.6 × 10^5^) and macrophages (SSC-H > 3.6 × 10^5^) (Appendix A). More than 85% of monocytes and macrophages were strongly stained for both CD14 and CD16 with >80% of CD14+ and CD16+ cells (Appendix A). The ratio of monocytes vs. macrophages was approximately 6:1. These two sub-populations displayed similar cell distributions in CD14 and CD16 staining plots. 

### 3.2. Analysis of Infected Cells by Flow Cytometry

Adherent PBMC were infected with ASFV at a MOI of 0.5 HAD_50_. Analysis of the of adherent PBMC after overnight infection showed a monocytes/macrophages ratio of approximately 1:2 (Appendix A). Lymphocytes appeared to be EGFP-negative (EGFP-) cells. The percentage of EGFP-positive cells (39%) in macrophages were 4.6 times higher than that in monocytes (8.5%) (Appendix A). Both monocytes and macrophages contained approximately the same frequency of CD16− and EGFP− cells (20.4% and 18.4%, respectively) and CD16+ and EGFP+ cells (0.2% and 1.2%, respectively), while, and importantly, nearly all EGFP+ cells were CD16− cells in both monocytes and macrophages fractions, with a monocytes vs. macrophages ratio of approximately 1:2 (Appendix A). In this study, the percentages of infected cells were based on the gating that included monocytes and macrophages.

### 3.3. Effect of Different MOIs on the Percentage of Infected Cells

Flow cytometry results showed that infected cells increased seemingly in a linear fashion from 0.1% to 65% along with increased MOIs (from 0.01 to 1 HAD_50_) after overnight incubation (Figure 1). Results demonstrated that the infection rate reached a plateau after MOI greater than 1 HAD_50_. The increase of the MOI values from 1 to 4 resulted in an increase of infection by only approximately 5%. Therefore, two infection doses (MOI of 0.5 and 0.05 HAD_50_) were selected to further optimize the virus neutralization assays. 

### 3.4. Time Course of EGFP Signaling and Virus Release

To further establish the optimal conditions to perform ASFV infection to be used in the neutralization assay, the number of infected macrophages (MOI = 1 HAD_50_) were monitored by flow cytometry at 3 h intervals post infection. The fluorescent signal of EGFP was first detected at 9 hpi, reaching the signal full intensity at 13.5 hpi (data not shown).

This time course experiment to quantify release of matured ASFV particles into cell culture supernatants shows that the percentages of EGFP+ cells at 3, 6 and 9 hpi were 0.3%, 0.1% and 0.1%, respectively, which were very similar values to those obtained in mock infected cultures (0.1%). The percentages increased significantly to 8.6%, 8.3% and 7.0% at 13.5, 15 and 18 hpi, respectively. Then the percentages decreased to 1.9 % at 24 hpi (Figure 2). At 27 hpi, the infected cells sharply increased again to a level higher (11.5%) than that observed at 13.5 to 18 hpi. At 30 and 33 hpi, the percentages of infected cells decreased to 1.9% and 0.9%, respectively. Another experiment at 10, 11 and 12 hpi showed that the percentages of infected cells were very similar to those at 3 to 9 hpi. These results indicate that the mature ASFV was firstly released into the supernatant between 12 and 13.5 hpi, which happened to be the mid-point of the second peak observed at 27 hpi. This second peak of virus release probably was due to the second-wave infection caused by the virus released in the 12–13.5 hpi peak. Therefore, a period of 13.5 h post-infection was further used as the incubation time in this study. Infection with cell lysates prepared from freezing-and-thawing of infected cells showed that infectious intracellular virions existed at 10, 11 and 12 hpi, corroborating the results obtained in the time course experiment previously described.

### 3.5. Virus Neutralization

ASFV treated with hyperimmune serum displayed significantly reduced percentage of infected cells in monocyte and macrophage populations compared to the viruses treated with naïve serum as examples in Figure 3 showed that ~75% of ASFV were neutralized by immune serum compared to naive serum. During the protocol optimization, several methodological alternatives were evaluated. When the mixtures of ASFV and sera were not removed after one hour of infection, the ASFV-positive serum displayed neutralizing effects of 74.4% and 67.3% compared to the nonimmune serum or only cell culture medium, respectively. When the mixtures of ASFV and serum were removed after one hour of infection, the neutralizing effects of the positive serum slightly increased (to 78.6% and 68.2%, respectively) (Figure 4). The comparative differences in the neutralizing effects of immune serum with pre-immune serum or with culture medium were statistically significant (*p* = 0.026), whereas the difference between removing or not removing the inoculum from the cell cultures was not significant (*p* = 0.070). Then, the neutralization assay was tested with adherent PBMC prepared from different donor pigs and similar results were obtained. These results indicated that inoculum containing naive serum enhanced ASFV infection compared to those containing only the culture medium. Therefore, the assays were further performed with the inoculum containing the virus/serum mix with 2-fold serially diluted hyperimmune sera and using a pre-immune serum as the control for basal neutralization activity.

To determine the effect of dilutions on the ability of immune sera to neutralize virus infectivity, adherent PBMC were alternatively infected with two ASFV MOIs (0.5 and 0.05 HAD_50_ of ASFV in 50 μL) while confronted with two-fold serial dilutions (from 1/2 to 1/128) diluted with culture medium. For MOI at 0.5 HAD_50_, the neutralizing effect decreased as the dilution folds increased up to 16×, while further dilution of sera produced an increase in neutralizing activity (Figure 5, Appendix A). Similar results were obtained using MOI of 0.05, although neutralization activity decreased at a slower rate compared to MOI at 0.5 up to 1/32 dilution (Figure 5, Appendix A). All differences in the neutralizing activity between immune and pre-immune sera were statistically significant except for the immune sera at 1/8 and 1/16 dilutions tested with MOI at 0.05 HAD_50_ (*p* = 0.06 and 0.14, respectively). The experiment with MOI at 0.5 HAD_50_ was repeated using adherent PBMC prepared from another pig and similar results were obtained with the trend turning at the same serum dilution of 1/16. Another similar experiment was conducted using pre-immune serum as the diluent. As expected, the neutralization activity decreased as dilutions increased without a turning point when a pooled negative/pre-immune serum was used as the diluent (Appendix A). Therefore, infection with different MOI displayed different protection for each of the immune serum dilution tested except for the 128-fold (Figure 5) with higher MOIs yielding lower neutralizing effects. Additional dilutions (256- and 512-fold) of hyperimmune sera showed lower percentages of neutralization than that at 128-fold dilution in a repeated experiment. These results indicate that lower MOI produce higher neutralizing titers, and the titers could also be affected by the diluent used, such as culture medium or naïve serum. 

### 3.6. Intracellular and Extracellular Virions

To determine the effect of serum on the infectivity of extracellular and intracellular ASFV, these viruses were suspended in culture medium, naïve sera and hyperimmune sera and used as the inoculum for the infection. Extracellular ASFV suspended in naïve sera showed significantly higher percentages (29.0%) of EGFP+ monocytes and macrophages than the virus suspended in culture medium (21.0%) (*p* = 0.02) and intracellular ASFV in naïve sera (23.6%) (*p* = 0.03), while there were not significant differences between intracellular ASFV suspended in the culture medium and naïve sera and between intracellular and extracellular ASFV in medium (Table 1). The results indicated that the serum increased the infectivity of extracellular but not intracellular virions. Hyperimmune sera showed incomplete neutralization (74.0%) against both extracellular virions and intracellular ASFV (66.2%).

## 4. Discussion

In this study, a fast and simple flow cytometry–based ASFV neutralization assay was developed to test the effect of hyperimmune sera in adherent pig PBMC. This assay takes less than 24 h and does not require cell staining or manual cell counting and allows for using a large number of virions. This flow cytometry–based assay has the potential to be applied to any ASFV strains by using anti-ASFV antibody staining to detect the infected cells instead of EGFP-expressing recombinant viruses. The timing of ASFV release from infected cells may be different among strains, which must be determined to avoid the complication of secondary infection. 

Flow cytometry showed that nearly half of adherent PBMCs used in this virus neutralization assay were lymphocytes, and that the ratio of monocytes vs. macrophages decreased with culture time. Only ~70% of the monocyte/macrophage population were infected by ASFV, even at a high MOI of 4 HAD_50_, in our assay settings because ASFV infection–resistant monocytes made up one-third of the population. Macrophages were close to five times more susceptible to ASFV infection than monocytes, as in those reported earlier [22,23]. Infected cells expressed practically no CD16 on the cell membranes, which also agreed with published studies [23,30]. The percentages of infected cells reached a plateau at MOI greater than 1 HAD_50_, lower than expected, which was due to that adherent PBMC contained lymphocytes and ASFV infection–resistant monocytes. Adherent human PBMC also contained lymphocytes, which were mostly T cells that were adhesive predominantly via interacting with adherent macrophages [31]. Our results agree with the report based on nearly 90% of CD3+ (a T lymphocyte marker) cells in the lymphocyte population. If only macrophages were taken into the account, the MOI of 1 in the adherent PBMC was equivalent to an MOI of 3 in macrophages. The MOI of 3 is expected to have 95% of infected cells based on the Poisson distribution.

As expected, we observed the ASFV neutralizing effect of hyperimmune sera from pigs vaccinated and challenged as well as a remaining non-neutralizable virus fraction. The highest neutralizing percentage in our assay conditions was ~80% for undiluted samples, which was generally lower than those reported by Zsak et al. [12] and Gómez-Puertas and Escribano [17] but similar to the report by Pérez-Núñez et al. [20]. In contrast to the report [17], we found inverse correlation between MOI and neutralizing efficacy. The differences probably were due to different assay settings including cells, serum dilutions and viruses used in the assays. We did not conduct side-by-side comparisons between flow cytometry–based and plaque reduction–based neutralization assays due to limited available adherent pig PBMC. A similar flow cytometry–based vaccinia virus neutralization assay was found to be more accurate and 10- to 20-fold more sensitive than plaque reduction assays [19]. 

Regarding vaccinia virus, extracellular virions are more resistant to antibody neutralization than intracellular virions [32]. There are two types of extracellular vaccinia virions, extracellular enveloped virions (EEV) and cell-associated enveloped virions (CEV), and CEV are more resistant to antibody neutralization than EEV [33]. Our results in this study showed that both extracellular and intracellular virions are resistant to antibody neutralization. ASFV CEV were also reported [34]. ASFV-infected cells lysed at Day 4 after infection in vitro [35]. The procedures of short incubation time (2 h) after starting release of extracellular virions and the centrifugation and filtration of the harvested supernatants should significantly reduce the contamination of other virions in the preparation of EEV; however, intracellular virions might contain a small number of CEV even though cell debris was removed by centrifugation after the freeze-and-thaw procedure. Approximately 25% of total virions were extracellular ASFV virions at 48 hpi for a genotype I isolate, BA71V [36], whereas our experiment in this study showed that ~40% of virions were extracellular virions at 15 hpi. Unlike ASFV, EEV of most poxviruses are <1% of total virions [37].

Interestingly, our results showed that extracellular ASFV suspended in naïve sera had higher percentages of infected cells than those in the culture medium, suggesting that certain components in naïve sera can enhance ASFV infection. Our results are consistent with those for extracellular and intracellular virions of the vaccinia virus [38]. The results for the vaccinia virus have been further tested with anti-Axl antibody (Axl is a Gas6 binding membrane protein) and ANX5 (a PtdSer binding protein). It is well known that serum proteins such as Protein S and Gas6 enhance some enveloped-virus infections in macrophages [39,40]. Protein S and Gas6 are phosphatidylserine (PtdSer)–binding proteins that are recognized by TAM receptors (Tyro3, Axl and Mer) on macrophages during endocytosis [41]. Macrophages are the primary cells in clearing apoptotic cells because of their high expression of receptors for PtdSer or PtdSer binding proteins. Receptors binding to PtdSer on the envelopes of viruses or PtdSer binding proteins can enhance virus entry and infection in macrophages, which is called apoptotic mimicry [42]. Apoptotic cells exposed PtdSer on the outer leaflets of the plasma membrane due to inactivation of flippases (a membrane protein that moves PtdSer and phosphatidylethanolamine from the outer leaflet to the inner leaflet of the plasma membrane) by activated caspase 3, and exoplasmic PtdSer acts as an “eat me” signal for macrophages to engulf and clear dying cells [41]. 

ASFV infection induced apoptosis in infected macrophages [43] and in adapted Vero cells [44] and activated caspase 3 [36,43]. Inhibition of caspase 3 activity in early ASFV infection blocked the production of the extracellular virions but not total virions [36]. Removal of phosphatidylinositol (PtdIns) from ASFV particles decreased neutralization efficacy and vice versa [16]. PtdSer and PtdIns are anionic phospholipids, the uneven distribution of which causes membrane curvature [45]. These findings suggest that exposing PtdSer on the outer leaflets of plasma membrane is needed for ASFV budding. Macrophages are the primary targets of ASFV infection in pigs, suggesting that receptor-mediated endocytosis is the main mechanism of entry [46]. The mechanisms of ASFV cell tropism are unknown. ASFV infects pig macrophages via both clathrin-mediated endocytosis and macropinocytosis [47,48,49]. The presence of PtdSer in the ASFV envelope and its involvement in ASFV infection could explain some of the results presented here, including the incomplete neutralization and the reason that ASFV primarily infects macrophages. 

## 5. Conclusions

A flow cytometry–based ASFV neutralization assay was developed and tested in pig adherent PBMC. Nearly half of adherent PBMC were characterized as lymphocytes in addition to monocytes and macrophages. The ratio of monocytes vs. macrophages decreased as culture time increased. Macrophages were approximately five times more susceptible to ASFV infection than monocytes, and infected cells displayed no CD16 expression on cell membranes. Undiluted hyperimmune sera reduced ASFV infection in pig monocytes/macrophages by ~80%. Hypothesized PtdSer binding–based on enhanced ASFV infection by naïve sera could explain in part why ASFV cannot ever be completely neutralized with hyperimmune sera and why ASFV primarily infects macrophages. Our results also show that ASFV-neutralizing titers can be affected by MOI and diluents used in the assay. In addition, it appeared that pre-immune/naive sera are better diluents than culture medium for the assay. Serum proteins in naïve sera enhanced the infection of extracellular virions in macrophages, suggesting a novel mechanism for ASFV infection in macrophages, as observed in some enveloped viruses. Both the intracellular and extracellular virions could not be completely neutralized by the hyperimmune sera.

## Figures and Tables

**Figure 1 viruses-14-01249-f001:**
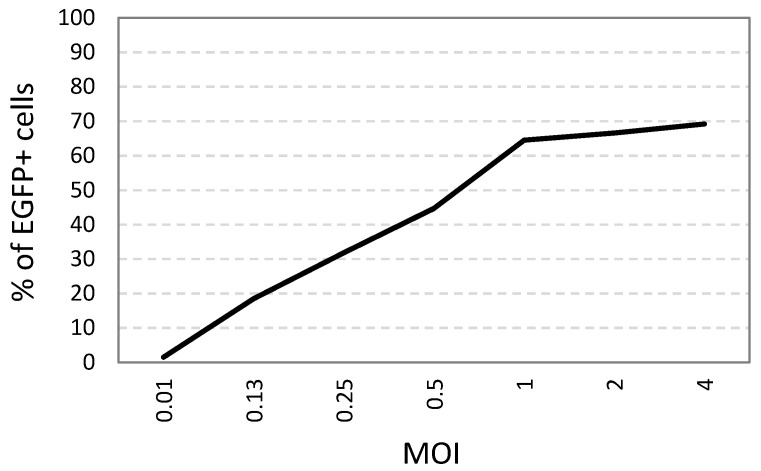
The percentages of ASFV-infected (EGFP+) cells at different MOI (HAD_50_) in cultured pig ex vivo adherent PBMC, calculated based on flow cytometry analysis gated on the population of monocytes and macrophages.

**Figure 2 viruses-14-01249-f002:**
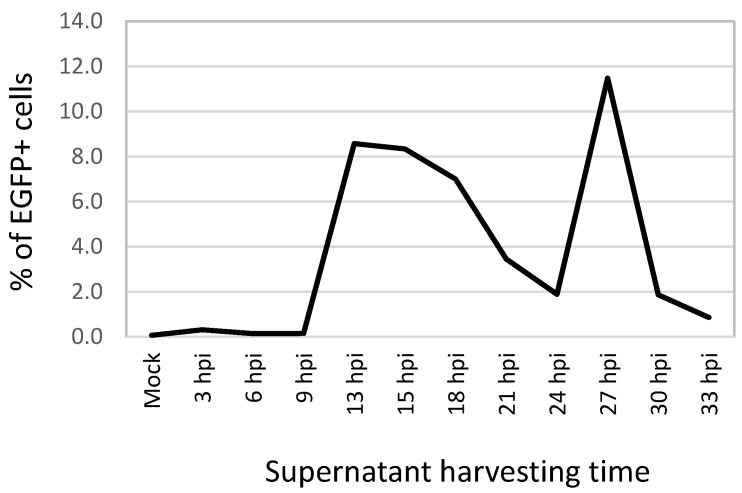
The percentages of EGFP+ monocytes and macrophages infected with 200 μL of supernatants harvested at different hours post-infection from pig ex vivo adherent PBMC cultured in a 12-well plate and infected with MOI of 1 HAD_50_ (the percentages were calculated based on flow cytometry analysis gated on the population containing monocytes and macrophages).

**Figure 3 viruses-14-01249-f003:**
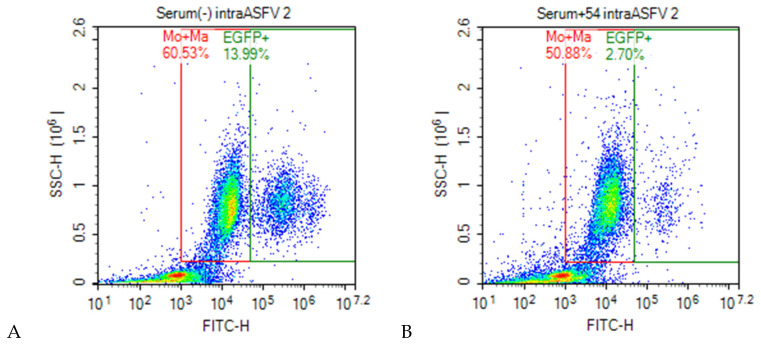
The cell distribution on the SSC vs. FITC (EGFP signal) plot of adherent PBMC gated on monocytes and macrophages (red) and EGFP+ cells (green) after infection with ASFV at MOI of 0.5 HAD_50_ treated with (**A**) naïve serum (−) and (**B**) hyperimmune serum (+).

**Figure 4 viruses-14-01249-f004:**
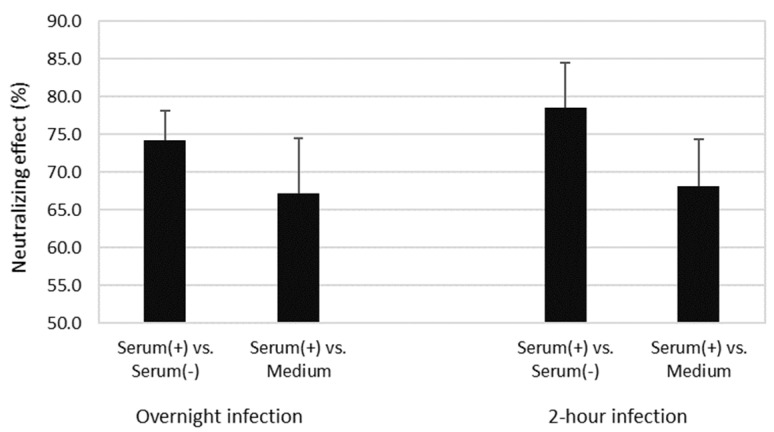
The neutralizing effects (percentage of EGFP+ cell reduction) of hyperimmunized sera compared to naïve sera or cell culture medium at MOI of 0.5 HAD_50_ (infection inoculum: 50 μL of ASFV in 250 μL of serum or medium) using adherent PBMC isolated from a donor pig.

**Figure 5 viruses-14-01249-f005:**
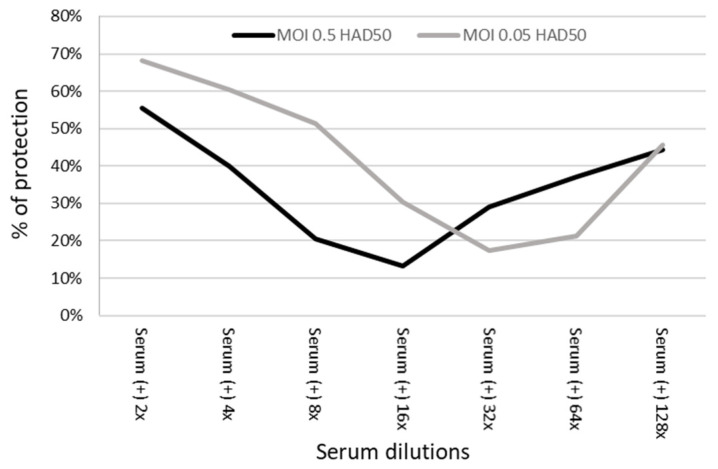
The neutralization expressed as the percentages of protection of 2-fold serially diluted positive sera compared to 2-fold diluted negative sera [protection % = (PI^−^ − PI^+^)/PI^−^, where PI^−^ is the percentage of infected cells treated with negative serum and PI^+^ is the percentage of infected cells treated with serum (−)] in gated monocytes and macrophages infected with different MOI.

**Table 1 viruses-14-01249-t001:** The percentages of EGFP+ monocytes and macrophages after infection with intracellular and extracellular ASFV suspended in culture medium and naïve sera (−) at MOI of 0.5 HAD_50._

	Medium	Serum (−)
Intracellular ASFV	22.2 ± 2.3	23.6 ± 1.1 ^a^
Extracellular ASFV	21.0 ± 1.7 ^b^	29.0 ± 5.2 ^a,b^

^a,b^ The differences between the treatments with the same superscripted letters were statistically significant (*p* < 0.05).

## Data Availability

Original data and details of experimental protocols are available upon request.

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
