# Peer review of "Serum Neutralizing and Enhancing Effects on African Swine Fever Virus Infectivity in Adherent Pig PBMC"

_viruses, 2022, doi:10.3390/v14061249_

Round 1

Reviewer 1 Report

The manuscript by Canter et al., presents an improved flow cytometry method to measure neutralization of African swine fever virus. This method uses an ASFV expressing eGFP for ease of detection but has potential to be adapted to other strains by using antibodies against ASFV proteins instead. The method presented improves on methods based on plaque reduction assays which take several days for assay and also improves he sensitivity of the assay. The authors carried out a range of optimizations based on moi used for infection, length of infection and dilution of the sera. They also investigated effects of infection for different times and in the presence of absence of sera. The results obtained are generally consistent with previously published results revealing a proportion of ASFV that cannot be neutralised.  The use of FACS allows the authors to investigate the proportion of cell populations in the PBMCs used for infection showing that the proportion of macrophages infected are higher than monocytes. Thus the neutralisation efficiency can be calculated based on proportion of susceptible cells. This may vary between PBMC cell populations from different pigs.

The current study includes several novel results. One of these was the comparison between neutralisation of extracellular and intracellular ASFV.  Extracellular but not intracellular virions suspended in naïve serum were more infectious than those in culture medium. The authors hypothesis that this may be due to binding of phosphatidyl serine, incorporated into the virus extracellular envelope, to phosphatidylserine receptors on infected macrophages. This may provide an alternative virus entry assay that is resistant to neutralising antibodies.  This is an interesting hypothesis although not further evaluated in this manuscript.

The results are generally well described and conclusions supported by the data. Some additional background information would help non-specialist readers to follow the manuscript.

Specific points:

  1. Lines 92, 93 The authors state “Serum samples were not heat-inactivated to include the effect of complements in the assay”. Discussion should be included to explain the possible impact of complements and how this may affect the conclusions drawn.
  2. The authors should include more description of the structure of ASFV and in particular explain the differences between extracellular and intracellular virions including some information about key proteins. Presumably since these forms were harvested from cell supernatants and cell lysates respectively without further purification there is likely to be contamination of the different fractions?
  3. The authors should explain what is known of the cell entry mechanisms for ASFV and proteins thought to be important in the process.

Author Response

     We are grateful and greatly appreciate the reviewer’s comments and suggestions, which have been taken into consideration in revising the manuscript. Changes based on these comments and suggestions significantly improve the manuscript.

Response to Comments and Suggestions

              We strongly agree with this reviewer that additional background information would help non-specialist readers to follow the manuscript. There are a lot of detail mechanisms involved in apoptotic mimicry and caspase 3-dependent ASFV extracellular virion release. The last paragraph in the discussion has been separated into two to include more background knowledge (see Lines 359-389 for details).

Response to Specific point 1:

Because antibodies have been demonstrated to kill ASFV-infected cells via complement-mediated cytotoxicity (CDC) (Norley and Wardley, 1982) and ASFV is known to not be completely neutralized by antibodies/immune sera, we decided not to heat-inactivate the serum samples to test if the complements could provide additional effect on ASFV infection. However, we did test both removal (2-hour infection, without CDC effect) and non-removal (over-night infection, possible CDC effect) of inoculum (the mixture of ASFV and immune serum) to compare the possible complement effect. Our results showed that ASFV still could not be completely neutralized even without removal of immune sera. The sentence “The complement effect was compared between removal and non-removal of inoculum (the mixture of ASFV and immune serum) as described later.” Has been added after Line 93 (see Lines 95-98).

Response to Specific point 2:

This is a very good point. Like poxviruses, there are three types of infectious ASFV virions, extracellular enveloped, cell-associated enveloped (reported by Jouvenet et al. in 2004 and 2006), and intracellular mature virions. The culture supernatants were harvested at 15 hours post-infection, which was 2 hours after starting release of extracellular virions [ASFV-infected cells lysed at Day 4 after infection in-vitro (Breese and DeBoer, 1966. 28:420-8)]. We did concern about the possible contamination in extracellular virions by intracellular virions; therefore, the supernatants were centrifugated and filtered through 0.5 μm syringe filters to remove infected cells. The harvesting time (short incubation), centrifugation and filtration should significantly reduce contamination of intracellular virions in extracellular virions. A sentence “The harvested supernatants were centrifugated and filtered through 0.5 μm syringe filters to remove infected cells before storage in the -70°C freezer” was added in the paragraph of Materials and Methods section (see Lines 123-125). The results of reference [35] have been included and cited in the Discussion section (see Lines 349-354).  However, the preparation of intracellular virions did contain cell-associated enveloped virions as pointed out in the discussion.

Response to Specific point 3:

              This is another excellent point. These two sentences “Macrophages are the primary targets of ASFV infection in pigs, suggesting that receptor-mediated endocytosis is the main mechanism of entry [46]. ASFV infects pig macrophages via both clathrin-mediated endocytosis and macropinocytosis [47,48,49].” Have been added in the discussion (Lines 384-387).

Reviewer 2 Report

The manuscript by Canter et al., describes studies to develop an improved method for determining neutralization of African Swine Fever virus (ASFV). Currently, the immune mechanisms involved in combatting virus this are not very clear but seem to involve both humoral and cellular systems. It has been difficult to develop effective vaccines against this virus. The current study aimed to use a GFP-tagged derivative of the virus in conjunction with flow cytometry to assess virus neutralization in a cell culture system (using PBMCs). This system has some advantages over conventional virus neutralization assays that use very low levels of virus.  

Major points:

  • The manuscript has been assembled in a rather odd manner. The first Figure mentioned is Figure 3A (line 184) and Figure 1A is then mentioned on line 192 and Figure 2 on line 199. The actual Figures do not appear until much later (after line 285), (c.f. Instructions to authors).
  • Figures 1-5 are essentially characterization of the system and do not show any data about virus neutralization. Personally, I think most of this information (or all) could be moved to Supplementary Information. In contrast, I lack seeing the comparison of the unprocessed cytometry results in the presence of positive and negative serum. I think this is a serious omission since it is key to the manuscript. The data for the virus neutralization is only presented in graphs (see Figures 6 and 7) and, unfortunately, there is no indication of the reproducibility of these assays, how many times were they performed? What is the extent of variation within the assays? This is a criticism of conventional virus neutralization assays and it would be good to demonstrate a higher level of consistency for this new assay. It can be expected that the authors have at least some of the necessary information but it should be presented here to make a more thorough analysis.

Other points:

  1. Considering this manuscript is produced from the US, there are numerous errors in the use of English that should be corrected to make the manuscript clearer. The sentence on lines 130-131 does not make sense to me.
  2. Some reference citations are inconsistent, e.g. Borca et al., 2017 is cited on line 99 without a number (maybe [24] and I think the “25” on line 144 is referring to reference [25] but this is not correctly formatted.
  3. In some places, “u” is used instead of “μ” (e.g. line 128, 156, 157).

Author Response

    We are grateful and greatly appreciate the reviewer’s comments and suggestions, which have been taken into consideration in revising the manuscript. Changes based on these comment and suggestions significantly improve the manuscript. The reviewer paid attention to details in reviewing the manuscript and caught several our careless mistakes, which are greatly appreciated.

Response to Major point 1:

  1. Figure 3A and 3B were changed to Figure 1. Thanks for catching these careless errors. Figures 1, 2, and 3 was changed to Supplemental Figures as suggested. Other figures were changed accordingly. We did not change Figures 4 and 5 to supplemental figures because these figures contain important optimization information of flow cytometry-based neutralization assay and new information of ASFV life cycle time.
  2. Unprocessed cytometry results in the presence of positive and negative serum were added as Figure 5. A sentence (Lines 251-253) was added to describe the figure. This is an excellent suggestion.
  3. Four immune serum samples from individual pigs were used in our experiments, each with duplicates in the assays. Changes have been made to clear this in Section 2.6 (see Lines 179-182).
  4. The variations (STD) of the neutralization assays have been included in the table or figures. We did not do side-by-side comparisons between conventional and flow cytometry-based assay; however, similar comparisons has been done for vaccinia virus neutralization assay as in Ref. 19, showing that flow cytometry-based assay was more accurate and 10- to 20-fold more sensitive than plaque reduction assay.

Response to Other points:

  1. The sentence on lines 130-131 has been revised. How MOI was calculated has been added to the paragraph (see Lines 138-144).
  2. Borca et al., 2017 was changed to Ref. [24]. The sentence with Ref. 25 was revised to make it clearer. Thanks for catching another careless mistake.
  3. Three “u”s have been changed to “μ” as suggested.
